# Pharmacological Small Molecules against Prostate Cancer by Enhancing Function of Death Receptor 5

**DOI:** 10.3390/ph15081029

**Published:** 2022-08-21

**Authors:** Xia Gan, Yonghong Liu, Xueni Wang

**Affiliations:** 1Guangxi Zhuang Yao Medicine Center of Engineering and Technology, Guangxi University of Chinese Medicine, 13 Wuhe Road, Qingxiu District, Nanning 530200, China; 2Institute of Marine Drugs, Guangxi University of Chinese Medicine, 13 Wuhe Road, Qingxiu District, Nanning 530200, China; 3CAS Key Laboratory of Tropical Marine Bio-Resources and Ecology, Guangdong Key Laboratory of Marine Materia Medica, South China Sea Institute of Oceanology, Chinese Academy of Sciences, Guangzhou 510301, China

**Keywords:** death receptor 5, prostate cancer, chemotherapeutic agents, natural compounds, synthesized compounds

## Abstract

Death receptor 5 (DR5) is a membrane protein that mediates exogenous apoptosis. Based on its function, it is considered to be a target for the treatment of cancers including prostate cancer. It is encouraging to note that a number of drugs targeting DR5 are now progressing to different stages of clinical trial studies. We collected 38 active compounds that could produce anti-prostate-cancer effects by modulating DR5, 28 of which were natural compounds and 10 of which were synthetic compounds. In addition, 6 clinically used chemotherapeutic agents have also been shown to promote DR5 expression and thus exert apoptosis-inducing effects in prostate cancer cells. These compounds promote the expression of DR5, thereby enhancing its function in inducing apoptosis. When these compounds were used in combination with the natural ligand of DR5, the number of apoptotic cells was significantly increased. These compounds are all promising for development as anti-prostate-cancer drugs, while most of these compounds are currently being evaluated for their anti-prostate-cancer effects at the cellular level and in animal studies. A great deal of more in-depth research is needed to evaluate whether they can be developed as drugs. We collected literature reports on small molecules against prostate cancer through modulation of DR5 to understand the current dynamics in this field and to evaluate the prospects of small molecules against prostate cancer through modulation of DR5.

## 1. Introduction

Prostate cancer is the second most common cancer in men in the world [1]. Androgen and androgen receptor play a critical role in the occurrence and development of prostate cancer [2,3]. Thus, androgen deprivation therapy (ADT) is the basic treatment for prostate cancer [4]. However, most patients inevitably develop into a more dangerous castration-resistant prostate cancer (CRPC) within 2 to 3 years after ADT treatment [4]. Chemotherapy is an important part of anti-prostate-cancer therapies [5]. Induction of apoptosis is a mechanism of action of chemotherapeutic agents in their anti-tumor process. Apoptosis is a process of programmed cell death, which occurs through exogenous apoptotic pathways (death-receptor-mediated) and endogenous apoptotic pathways (mitochondria-mediated) [6]. The extracellular apoptosis pathway is initiated by death receptors on the cell surface, such as Fas and the tumor necrosis factor receptor family, which activate intracellular apoptotic enzyme caspase through extracellular signals to induce cell death [7].

Death Receptor 5 (DR5) is also known as tumor-necrosis-factor (TNF)-related apoptosis-inducing ligand receptor 2 (TRAIL-R2). DR5 is made up of 411 amino acids and it contains the death structure domain, a part that is located inside the cell [6]. When the ligand and DR5 bind as a trimer, the death structural domain of the activated receptor binds to the Fas-associated death domain (FADD) to induce apoptosis. The FADD contains two interacting structural domains, the death domain (DD) and the death effector domain (DED) [7]. FADD acts as a bridge between DR5 and multiple procaspase-8 molecules, which bind to the death effector domain (DED) of FADD to form the death-inducing signaling complex (DISC). Then, two procaspase-8 activated in DISC undergo intramolecular hydrolysis and subsequently dimerize to form caspase-8, thereby initiating the cellular apoptotic process [7]. DR5 is a target for inducing cancer cell apoptosis by chemotherapy [8]. DR5 is widely expressed on the surface of tumor cells, but rarely on normal cells [9,10,11]. Based on its differential expression in cells, DR5 is a potential target for tumor therapy with higher efficacy and safety. However, the expression of death receptor 5 was inversely correlated with the progression of prostate cancer [12]. Therefore, increasing the expression of DR5 in prostate cancer cells is required to achieve induction of apoptosis via DR5.

Agonists of death receptors are considered as a strategy for targeted cancer therapy. Currently known DR5 agonists include antibodies, proteins, and small molecule compounds. Several anti-DR5 antibodies have been tested in clinical trials; however, the results of randomized phase II studies have failed to show significant clinical benefit [13]. Circularly permuted TRAIL and dulanermin are two representative recombinant protein DR5 ligands, of which circularly permuted TRAIL has been applied for marketing in China for the treatment of multiple myeloma. Dulanermin has also been in a phase III clinical trial for advanced non-small-cell lung cancer [14]. Smaller molecules of DR5 agonists have been reported less frequently. Bioymifi is the first known small molecule agonist of DR5, and Wang et al. found that it can directly bind to DR5 to drive apoptosis. They also demonstrated the anti-tumor effects of bioymifi in glioma cells [15]. However, these DR5 agonists that have been tested in clinical trials are not yet indicated for the treatment of prostate cancer. In other words, there are currently no DR5 agonists, including antibodies, proteins, and small molecule compounds, in clinical trials for prostate cancer.

In this review, our literature search was performed using the PubMed database up to 30 July 2022. The search was conducted using the keywords “(prostate cancer (Title)) AND (DR5 (Title)) “ and “(prostate cancer) and (death receptor 5)”. We found that studies related to both DR5 and prostate cancer began in 1999, with 216 papers reporting this content to date. We screened and excluded irrelevant papers by title, abstract, and content, and obtained 40 papers that met the requirements, which were focused on small molecule compounds regulating apoptosis in prostate cancer cells via DR5. Large molecules that regulate DR5, small molecule compounds that do not alter DR5 expression, and small molecule compounds that regulate DR5 but do not affect prostate cancer are not the object of the present review. In this study, our literature research revealed that a variety of small molecules have been reported to regulate the DR5 signaling pathway to induce apoptosis. Interestingly, however, these small molecules do not directly bind to DR5 to activate it to induce apoptosis, but rather promote DR5 expression in various indirect ways to enhance its function. Some of these small molecules are natural and some are chemically synthesized. Here, we collect and classify these compounds, observe the stage at which they are involved in anti-prostate-cancer research, summarize their anti-prostate-cancer characteristics, and finally predict the potential of these compounds to develop into anti-prostate-cancer drugs.

## 2. Chemotherapeutic Agents That Modulate the Death Receptor 5 Signaling Pathway

Paclitaxel, vincristine, vinblastine, etoposide, doxorubicin, and camptothecin are all anticancer chemotherapy drugs. These chemotherapeutic agents were found to have the effect of upregulating DR5 expression, thereby promoting the action of TRAIL and increasing the rate of apoptotic cell death [16]. Animal studies have shown that the combination of these chemotherapeutic agents and TRAIL can effectively enhance the anti-prostate-cancer effects of both alone [17].

The six chemotherapy drugs mentioned above are not commonly used in the clinical treatment of prostate cancer. However, with the exception of vincristine and camptothecin, the other four chemotherapeutic agents have been tested in clinical trials against prostate cancer with positive results. Kelly et al. [18] conducted a multicenter phase II trial in patients with untreated limited prostate cancer to assess the safety and feasibility of radiotherapy following paclitaxel, estradiol phosphate, and carboplatin plus androgen deprivation therapy. The results showed that the therapy did not cause grade 4 acute and late genitourinary toxicity, but there were common grade 1 to 2 late toxicities such as proctitis, dyspareunia, urinary frequency, and urgency. Millikan et al. [19] conducted a randomized, multicenter phase II clinical trial of two multi-component regimens in the treatment of androgen-dependent prostate cancer in patients with progressive and androgen-dependent prostate cancer. Patients were treated alternately with ketoconazole/doxorubicin and vincristine/estimostatin or paclitaxel, estimostatin, and oral etoposide. Patients were then prospectively stratified according to the volume of disease and observed for prostatitis. Results showed that 30% of patients had decreased prostate-specific antigen and prolonged median survival [19]. Laber et al. conducted a phase I and phase II clinical trial with adriamycin in combination with docetaxel and found pain relief in 73% of prostate cancer patients, and only 3% had adverse effects such as neutropenia, stomatitis, headache, and weight loss [20].

## 3. Natural Compounds Inhibiting Prostate Cancer by Targeting Death Receptor 5

Natural compounds have long been one of the main sources for the discovery of anti-tumor drugs. Scientists have identified dozens of small natural molecules that can produce anti-prostate-cancer effects by modulating the DR5 signaling pathway.

Acetyl-keto-β-boswellic acid (AKBA) is a triterpene isolated from birch and serrated trees [21]. Lu et al. [22,23] found that AKBA inhibited the apoptosis of LNCaP and PC-3 cells, thereby suppressing their proliferation. AKBA-induced apoptosis was initiated by activation of caspase-8, which was driven by DR5. This was demonstrated by the inhibition of caspase-8 activation when the death receptor was knocked down by small hairpin RNAs. Moreover, AKBA treatment of prostate cancer cells increased the expression level of the CAAT/enhancer-binding protein homologous protein (CHOP) and activated the DR5 promoter reporter gene. Briefly, AKBA induces apoptosis in prostate cancer cells through a DR5-mediated pathway, which is associated with the induction of CHOP expression.

Apigenin is a common flavonoid that is expressed in many plants, such as the rubenaceae family [24], verbenaceae family, and sellopressaceae family [25]. Oishi et al. found that apigenin could induce apoptosis of prostate cancer DU145 cells by binding to adenine nucleotide translocase-2 (ANT2) and inhibiting the activity of ANT2 and then upregulating DR5 expression. This study suggests that apigenin can promote DR5 expression to induce apoptosis in prostate cancer cells [26].

Artemisinin C is a bioactive phenolic component extracted from *Baccharis dracun-culiforia* [27]; it has a variety of biological activities, including anti-oxidant, anti-bacterial, anti-inflammatory, anti-genotoxic, anti-angiogenic, and anti-cancer properties and so on [28]. Szliszka et al. [29] treated TRAIL-resistant LNCaP prostate cancer cells with TRAIL and artemisinin C, then analyzed the expression of death receptors (TRAIL-R1/DR4 and TRAIL-R1/DR5) by flow cytometry and determined the activity of caspase-8 and caspase-3 by trypsin assay; they found that artemisinin C sensitized TRAIL-resistant LNCaP cells by participating in exogenous (receptor-mediated) and endogenous (mitochondrial) apoptosis pathways. Artemisinin C increased the expression of DR5 and decreased the activity of NF-κB. TRAIL in combination with artemisinin C significantly induced activation of caspase-8 and caspase-3, as well as mitochondrial membrane potential (∆Ψm) [29]. These results confirm that artemisinin C plays a role in cancer chemoprevention by regulating the TRAIL-mediated apoptosis signaling pathway.

Auriculasin is an allylated isoflavone, isolated from the root of *Flemingia philippinensis*, with antitumor activity [30,31] RC-58T/h/SA#4 is a primary prostate cancer cell line and RWPE-1 is a normal prostate epithelial cell line. In Cho et al.’s study, auriculasin induced apoptosis in RC-58T/h/SA#4 cells while showed no cytotoxicity in RWPE-1 cells. In RC-58T/h/SA#4 cells, combined use of auriculasin and TRAIL at optimal concentrations resulted in specific apoptotic cell death accompanied by increased expression of DR5 and pro-apoptotic proteins. Simultaneously, the authors found that auriculasin could enhance the transcriptional activity of the DR5 transcription factor CHOP, which supports the idea that auriculasin induces apoptosis through upregulation of DR5 expression [32].

Baicalein is a flavonoid extracted from the roots of *Scutellaria* with antibacterial, antiviral, antioxidant, and pro-oxidation effects [33,34]. Baicalein combined with TRAIL could increase the apoptosis sensitivity of PC-3 cells and effectively increase the apoptosis induced by TRAIL and baicalein alone. Further studies showed that baicalein upregulated DR5 expression at the mRNA and protein levels, as well as that baicalein induced reactive oxygen species (ROS) expression. In contrast, ROS scavengers inhibited DR5 expression and TRAIL sensitization in PC3 cells. In conclusion, baicalin induces apoptosis of cancer cells by upregulating the expression of DR5, which may be related to the activation of ROS [35].

Biochanin-A, which has anti-cancer and chemopreventive properties, is the main dietary isoflavone found in soy and red clover [36]. LNCaP and DU145 prostate cancer cells are known to be resistant to TRAIL-induced apoptosis. Szliszka et al. [37] used the LNCaP cell line to study the mechanism of biochanin-A-enhancing TRAIL-mediated apoptosis. They examined the effect of biochanin-A on the expression of DR5 on LNCaP cells by flow cytometry. The results showed that biochanin-A increased the expression of DR5. In order to confirm that the ability of biochanin-A combined with TRAIL to induce apoptosis is mediated by DR5, they used a TRAIL-R2/FC chimeric protein which has a dominant negative function on TRAIL-R2. The results showed that the protein effectively blocked the apoptosis caused by the cotreatment of biochanin-A and TRAIL. These data suggest that biochanin-A could inhibit the proliferation of prostate cancer cells and mediate programmed cell death. In conclusion, biochanin-A enhances the sensitivity of prostate cancer cells to TRAIL and induces apoptosis of prostate cancer LNCaP cells by regulating the expression of DR5.

Cordycepin is an active component of the *fungus cordyceps*, with strong antioxidant and anticancer activities [38,39]. Lee et al. studied the effect of cordycepin on prostate cancer cells and found that cordycepin could induce LNCaP cell death. RT-PCR and Western blot assays showed that cordycepin increased the expression of DR5 in LNCaP cells at both the mRNA and protein levels. In short, cordycepin induces apoptosis of LNCaP by increasing DR5 expression at the protein and mRNA levels [40].

Cryptocaryone is a natural dihydrochaldone and it is an effective anticancer agent isolated from *Cryptocarya infectoria* [41]. Cryptocaryone increased the expression of the Fas protein in lipid rafts, which led to FADD, caspase-8, death receptor 4 and DR5 also being involved in this event. Confocal immunofluorescence assay confirmed that cryptocaryone triggered the aggregation of death receptors and recruited FADD and procaspase-8, resulting in the activation of caspase-8 and caspase-3 and apoptosis in prostate cancer cells. In conclusion, cryptocaryone exerts its anticancer activity by stimulating death receptors and related molecular clusters, and then induces prostate cancer cell apoptosis.

Delphinidin is a polyphenol compound with anti-inflammatory, antioxidant, and antitumor activities [42]. Delphinidin inhibits cell proliferation and induces cell apoptosis in many different cancer models, including colon cancer, uterine cancer, breast cancer, and prostate cancer [43]. Hyeonseok et al. [43] studied the effect of delphinidin on prostate cancer cells. They found that delphinidin sensitized prostate cancer cells to TRAIL-induced apoptosis by activating DR5 and caspase-mediated cleavage of histonedeacetylase 3 (HDAC3). Delphinidin increased DR5 protein levels in DU145 and LNCaP prostate cancer cells. When DR5 was knocked down by SiRNA, the apoptosis induced by delphinidin and TRAIL was inhibited, confirming the functional significance of DR5 in the process of delphinidin-stimulated TRAIL-mediated apoptosis.

Diallyl trisulfides (DATs) are the main organic sulfur compounds in garlic and are widely used as food flavorings. DATs can promote the activation of T cells and enhance the anti-tumor function of macrophages; these findings suggest a potential application of DATs in anti-tumor therapy [44]. The effects of DATs on prostate cancer cells in vivo and in vitro were investigated by Shankar et al. They found that DATs inhibited cell viability and cell clonal colony formation in PC-3 and LNCaP cells and induced apoptosis in these cells. DATs induced the expression of DR4 and DR5, which induced apoptosis in LNCaP cells, leading to their proliferation being inhibited [45]. The combination of DATs and TRAIL is more effective in inhibiting prostate tumor growth than each of them alone. The combination of them increased the expression of DR4 and DR5, activated caspase-8, and induced apoptosis. In short, DATs induced apoptosis of LNCaP and PC-3 cells through the regulation of death receptor 4/5 expression.

Ergosterol peroxide is derived from Neungyi mushrooms (*Sarcodon aspratus*) and it has immunomodulatory, anti-inflammatory, and anticancer effects [46,47]. Han et al. found that ergosterol peroxide had a strong cytotoxic effect on DU145, PC-3, M2182 and other prostate cancer cell lines in a concentration-dependent manner, and the apoptosis induced by ergosterol peroxide was DR5 dependent. In particular, ergosterol peroxide has a significant cytotoxic effect on DU145 prostate cancer cells; it increases the sub-G1 population and TUNEL-positive cells and activates DR5. In conclusion, these findings suggest that activation of DR5 plays a key role in ergosterol peroxidation inducing apoptosis in DU145 prostate cancer cells [48].

Flavokawain B is a kava chalcone with anticancer activity [49]. Tang et al. treated DU145, PC-3, LAPC4, and LNCaP cells with flavokawain B and found that flavokawain B induced apoptosis in cancer cells but not normal prostate epithelial cells. Furthermore, flavokawain B enhanced apoptosis induced by TRAIL through increasing the mRNA and protein expression of DR5 [50].

Indole-3-methanolis a phytochemical produced in fruits and vegetables; it can prevent many types of cancer, including hormone-related cancers [51]. Jeon et al. treated LNCaP cells with indole-3-methanol before TRAIL and found that indole-3-methanol enhanced TRAIL-mediated apoptosis and the expression of DR5 at the transcriptional and translational levels. In summary, the sensitization effect of indole-3-methanol on TRAIL is related to the upregulation of DR5 at both the mRNA and protein levels [52].

Isosilybin A has been identified as one of the bioactive components of silymarin and has a strong tendency to induce cell death in PCa cells [53]. Gagandeep et al. treated three different human PCa cell lines, 22Rv1, LAPC, and LNCaP, with isosilybin A. Both exogenous apoptotic pathways (increased levels of DR5 and cleavage of caspase 8) and endogenous apoptotic pathways (activation of caspases 9 and 3) were induced, resulting in apoptotic cell death [54].

Nordihydroguaiaretic acid (NDGA) is a natural substance isolated from the creosote bush (*larra triedentata*) [55]. Traditionally, NDGA has been regarded as a specific and most powerful inhibitor of lipoxygenase in arachidonic acid metabolism with anticancer activity [56]. Yoshida et al. [57] found that NDGA enhanced TRAIL-induced apoptosis of DU145 cells in a dose-dependent manner by studying the effect of NDGA on prostate cancer. The results of Western blot and silencing siRNA showed that NDGA upregulated the expression of DR5 in DU145 cells at the mRNA and protein levels, thereby inducing apoptosis of prostate cancer DU145 cells. In conclusion, NDGA induces apoptosis of prostate cancer DU145 cells by upregulating the expression of DR5 and activating caspase-mediated TRAIL. Furthermore, Nordihydroguaiaretic acid has progressed to Phase Ⅱ clinical trials in anti-prostate-cancer studies, suggesting that it is a promising anti-prostate-cancer drug [58].

Ouabain is a cardiotonic steroid extracted from natural plants and has the potential to induce apoptosis of cancer cells [59]. Chang et al. [60] found that ouabain increased the expression of DNA-damage-related proteins and proteins associated with exogenous apoptotic pathways (DR4, DR5, Fas, and FADD). In their study, ouabain reduced the survival of DU145 cells in a dose-dependent manner and activated the function of a series of proteins associated with apoptosis (Fas, TRAIL, DR5, DR4, caspase-8, and caspase-3), thereby inducing apoptosis.

Quercetin is a bioactive plant flavonoid with antioxidant, anti-inflammatory, and anticancer activities [61]. Jung et al. [62] found that quercetin enhanced TRAIL-induced apoptosis in prostate cancer cells by improving the protein stability of DR5. Quercetin increased DR5 expression in prostate cancer cells in a dose-dependent manner, which was mediated by increased transcription and protein stability. Quercetin induced DR5 expression at the transcriptional and translational levels and increased the level of DR5 protein in prostate cancer cells. The increase in DR5 expression leads to the stimulation of the death receptor pathway and the activation of caspase-8, which induces the apoptosis of cancer cells [62].

Resveratrol is a natural polyphenol compound that can be used to prevent and treat a variety of human cancers. Chen et al. [63] used prostate cancer LNCaP cells to study the molecular mechanism of resveratrol-induced apoptosis. The results showed that resveratrol could inhibit the phosphorylation of PI3K, Akt, and mTOR, then induce FOXO transcriptional activity, upregulate the expression of DR4 and DR5, inhibit the proliferation of cancer cells, and induce apoptosis. In addition, resveratrol can also upregulate the expression of death receptor 4 and DR5 and induce apoptosis by producing reactive oxygen species, targeting mitochondrial p53 and regulating Bcl-2 family members [64].

Retigeric acid B is a natural pentacyclic triterpenoid acid isolated from the *Lobaria kurokawae* Yoshim with significant antitumor activity [65,66]. Liu et al. evaluated the effect of retigeric acid B combined with clinical chemotherapy drugs on PC-3, DU145, and RWPE-1 cells. They found that low-dose retigeric acid B combined with cisplatin had significant synergistic cytotoxicity. Moreover, retigeric acid B activated the pro-apoptotic protein DR5 and enhanced the chemotherapy response of cisplatin. When DR5 was knocked down it would partially block retigeric acid B-cisplatin synergy, suggesting that DR5 plays a crucial role in this event [67].

Sulforaphane is a natural isothiocyanate with antioxidant, antiproliferative, and anti-cancer properties [68]. Shankar et al. [69] studied the therapeutic effect of sulforaphane on LNCaP and PC-3 cells through in vitro experiments and establishing xenotransplantation models. Their results showed that sulforaphane enhanced the therapeutic effect of TRAIL on PC-3 cells and sensitized TRAIL-resistant LNCaP cells. Sulforaphane can enhance the anti-tumor activity of TRAIL, upregulate the expression of DR5, and induce apoptosis of PC-3 cells.

Tanshinone I is the main bioactive compound of *Salvia miltiorrhiza* [70]. Eun Ah Shin et al. [71] found that the combination of tanshinone I and TRAIL could enhance the protein expression of DR5 (DR5) and weaken the expression of antiapoptotic protein. RT-PCR and RT-qPCR analysis confirmed that tanshinone I combined with TRAIL could upregulate DR5 and microRNA 135a-3p (miR 135a-3p) at the mRNA and DR5 promoter levels and reduce the phosphorylation of PC-3 extracellular-signal-regulated kinase. Interestingly, microRNA 135a-3p mimics enhanced DR5-mRNA and increased the number of PARP-cleavage-, Bax-, and TUNEL-positive cells in tanshinone I and TRAIL-cotreated PC-3. In conclusion, tanshinone I can induce the death of prostate cancer PC-3 and DU145 cells by upregulating the levels of mir135a-3p and DR5 in prostate cancer cells in combination with TRAIL.

Tetrandrine is a natural dibenzyl isoquinoline alkaloid isolated from the root of tetrandrine, which has a variety of pharmacological activities, including immunosuppressive, antihypertensive, and antitumor activities [72]. Tetrandrine upregulated mRNA expression and protein expression of death receptor 4 and DR5, causing TRAIL-resistant (LNCaP) and mildly TRAIL-sensitive (PC3) cells to respond to TRAIL-induced apoptosis. The critical requirement of DR4 and DR5 in TRAIL-sensitized PCa cells was validated by shRNA knockdown technology [73]. When PCa cells were treated with tetrandrine, the mRNA and protein levels of DR4 and DR5 would increase in a dose- and time-dependent manner [74].

Triptolide is a diterpene compound isolated from *Tripterygium wilfordii* [75]. It can inhibit the growth of and induce apoptosis of a variety of human tumor cells. Wen [76] found that triptolide was sensitive to TRAIL-induced apoptosis in prostate cancer cells through p53-mediated upregulation of DR5. Triptolide can inhibit the Akt/HDM2 signaling pathway, leading to p53 accumulation, and thus upregulate the expression of DR5, but not DR4. Tumor suppressor gene p53 is an important regulator of cancer cell apoptosis. DR5 is the first gene that directly activates its transcription through p53 [77]. Triptolide treatment can induce apoptosis of PC-3 and LNCaP cells by upregulating DR5 expression, dependent on p53.

Tunicamycin is a natural antibiotic [78]. Tunicamycin promoted TRAIL-induced apoptosis by upregulating expression of death receptor 4 (DR4) and DR5 while downregulating expression of apoptosis inhibitor 2 (clAP2). Tunicamycin significantly sensitized the androgen-independent prostate cancer cell line PC-3, making it sensitive to TRAIL-induced apoptosis. Tunicamycin upregulated DR5 expression at the mRNA and protein levels in a dose-dependent manner. In addition, the sensitization of TRAIL mediated by tunicamycin could be effectively reduced by DR5 small interfering RNA. Furthermore, tunicamycin also stimulated CHOP, the upstream regulator of DR5. Tunicamycin increased the activity of DR5 promoter, which was weakened by the mutation of the CHOP binding site [79,80].

Ursodeoxycholic acid is a hydrophobic bile acid with anticancer activity [81,82]. Ursodeoxycholic acid could significantly inhibit the growth of DU145 cells in a dose-dependent manner. Apoptosis induced by ursodeoxycholic acid is related to external pathways, evidenced by the fact that it triggered the exogenous pathway of apoptosis via upregulating DR5 expression. Then, it activated the endogenous pathway by regulating Bax, Bcl-XL, and cytochrome C release, inducing apoptosis in human DU145 prostate cancer cells [83].

Ursolic acid, which exists in the form of aglycones or glycosides in a variety of plants [84,85], is a pentyclic triterpenoid carboxylic acid with anti-tumorigenesis activity [86]. Shin and Park studied the effect and mechanism of action of ursolic acid on LNCaP, PC3, and DU145 prostate cancer cells; the results showed that ursolic acid could significantly promote TRAIL-induced apoptosis of prostate cancer cells and induce CHOP-dependent upregulation of DR5. Ursolic acid treatment leads to CHOP-mediated upregulation of DR5, making cancer cells more sensitive to the cytotoxic activity of TRAIL, improving the efficiency of TRAIL-induced apoptosis in cancer cells [86].

Vitamin A is a dimer of resveratrol and it has cytotoxic, anti-fat, anti-inflammatory, and antioxidant effects [87]. Shin et al. [88] found that vitamin A could increase the sensitivity of prostate cancer PC-3 cells to apoptosis by upregulating the level of DR5 and stimulating the production of reactive oxygen species (ROS). In PC-3 cells, vitamin A combined with TRAIL can reduce the expression of procaspase7/8 and activate caspase-3, FADD, DR5, and DR4. In addition, vitamin A combined with TRAIL can also upregulate the level of DR5 on the cell surface and induce apoptosis of cancer cells by increasing reactive oxygen species (ROS). The induction of DR5 protein expression in PC-3 cells by vitamin A and TRAIL significantly enhanced the DR5 promoter activity and DR5 expression on the cell surface and intensified the induction of apoptosis in cancer cells.

Xanthohumol is a natural flavonoid derivative with broad-spectrum antibacterial, antioxidant, anticancer, chemopreventive, and anti-inflammatory properties [89]. Kosekton et al. [90] found that the potential mechanism of apoptosis following TRAIL binding to xanthohumol might be related to activation of caspase-3, caspase-8, caspase-9, or bid; increased Bax expression; decreased bcl XL expression; and decreased mitochondrial potential in LNCaP cells. Xanthohumol is sensitive to TRAIL-induced apoptosis of LNCaP cancer cells; the expression of DR5 in LNCaP cells was upregulated after treatment with TRAIL and xanthohumol, promoting its programmed death.

## 4. Synthesized Compounds Inhibiting Prostate Cancer by Targeting Death Receptor 5

ABT-737 is a small molecule inhibitor of antiapoptotic protein Bcl-2 screened and modified from the chemical library by Oltersdorf et al. using nuclear magnetic resonance (NMR) screening, parallel synthesis, and structure-based design [91], which has a variety of anticancer activities, including against lymphoma, small-cell lung cancer, kidney cancer, prostate cancer, and lung cancer [92]. Song et al. [93] studied the effect of ABT-737 on prostate cancer cells through experiments. The results showed that ABT-737 combined with TRAIL increased the apoptosis sensitivity of DU145 cells. The apoptotic cascade components were detected by Western blot and their mechanism was studied. It was found that ABT-737 did not change the levels of FADD and caspase-8 but increased the level of TRAIL receptor DR5 by activating the DR5 promoter, increasing the transcription-factor enhancer-binding protein homologous protein (CHOP) and stimulating the increase in DR5 mRNA. Moreover, in prostate cancer cells, ABT-737 can also induce the production of reactive oxygen species (ROS), upregulate the expression of DR5, and induce apoptosis of DU145 and LNCaP cells [94].

Allopurinol is an inhibitor of xanthine oxidase [95]. Xanthine oxidase plays an important role in various inflammatory diseases and chronic heart failure. It also plays an important role in the treatment of cancer. Takashi Yasuda et al. found that the combination of allopurinol and TRAIL can significantly induce apoptosis of human hormone refractory prostate cancer cells. Allopurinol enhanced the activity of DR5 promoter in PC-3 and DU145 cells by inducing CAAT/enhancer-binding protein homologous protein (CHOP), increasing DR5 protein, upregulating DR5 expression, and inducing apoptosis in PC-3 and DU145 cells [96]. In conclusion, allopurinol increases the sensitivity of prostate cancer cells to TRAIL-induced apoptosis by transcriptional regulation of DR5 expression.

N, N’-[(3,4-dimethoxyphenyl) methylene] biscinnamide (DPMBC) is a synthetic compound with anti-tumor activity [97,98]. Hiromi et al. used DPMBC to treat PC-3 and found that DPMBC inhibited the proliferation of and induced apoptosis of PC-3 cells. Western blot revealed that DPMBC upregulated the expression of DR5 in PC-3 cells. In a word, DPMBC upregulated the expression of DR5, activated the exogenous apoptotic pathway, and induced apoptosis [97].

C25 is a new compound screened from a small molecular library based on its similarity to the redox activity of curcumin and its sensitivity to TRAIL-induced apoptosis [99]. It has anti-lung-cancer, -ovarian-cancer, -pancreatic-cancer and -prostate-cancer activity [100]. James et al. [99] measured the effect of C25 combined with TRAIL on human cancer cell lines through an apoptosis test and Western blot. Their results showed that C25 could upregulate DR5 expression and enhance TRAIL-mediated cell apoptosis. C25-induced DR5 expression was related to the increase in oxidative stress and the accumulation of CHOP (CHOP is a stress-regulated transcription factor that can directly activate the transcription of DR5). In short, C25 induces LNCaP cell apoptosis by upregulating DR5 expression.

Cyproterone acetate (CPA) is a synthetic steroid that was initially used in the treatment of prostate cancer because it could block androgen receptor (AR) and reduce serum testosterone levels [101]. Chen et al. [102] found that PC-3 and DU145 cells showed greatly improved sensitivity to TRAIL after being pretreated with CPA. Moreover, cyproterone acetate activated the DR5 promoter by stimulating the transcription factor CCAAT-enhancer-binding protein homologous protein (CHOP), thereby increasing the mRNA and protein expression of DR5 (DR5), and inducing apoptosis of PC-3 and DU145 cells. In conclusion, CPA promotes androgen independent prostate cancer cell apoptosis by upregulating DR5. Cyproterone acetate has completed Phase Ⅲ clinical trials in anti-prostate-cancer studies, indicating that it is a promising anti-prostate-cancer drug [103].

Dihydroartemisinin (DHA) is a derivative of artemisinin [104]. Studies showed that DHA inhibited the function of the PI3K/Akt pathway in PC3, LNCaP, and DU145 prostate cancer cells, triggered DR5, and then activated exogenous and endogenous cell death signals. DHA induces the expression of DR5 by increasing the transcriptional activity of the DR5 promoter and upregulates the level of DR5 protein. In addition, the combined use of DHA and TRAIL can significantly enhance the killing capacity of cells [105]. In conclusion, DHA increases the expression of DR5 gene by activating DR5-mRNA and promoter, upregulates the expression of DR5, and induces apoptosis of prostate cancer PC3, DU145 and LNCaP cells, but has little cytotoxicity to normal prostate epithelial cells.

Norcantharidin is a synthetic demethylated analogue of cantharidin that has a variety of anticancer activities, including against oral cancer, liver cancer, and prostate cancer [106]. Yang et al. [107] found that norcantharidin induced cytotoxicity in prostate cancer cells in a dose- and time-dependent manner. Norcantharidin essentially increased the level of cytosolic cytochrome C and activated caspase-3, caspase-8, and caspase-9. Norcantharidin upregulated the expression of Fas and DR5 in 22Rv1, which strongly indicated that exogenous pathway was involved in norcantharidin-induced PCa cell apoptosis.

Saquinavir-NO (Saq-NO) is a derivative of the HIV protease inhibitor saquinavir [108]. It is synthesized based on the chemical modification of adding a nitric oxide (NO) donor group. Its toxicity is different from that of the parent compound, and it has a non-toxic variant of the parent drug that enhances its anticancer activity. This new chemical shows strong cell inhibitory and anticancer effects in vitro and in vivo, and the dose is significantly lower than that of the parent compound [109]. Donia et al. [110] tested the effect of saq-NO on the growth of the human-androgen-independent PC-3 prostate cancer cell line in vitro and in vivo. The results clearly showed that saq-NO was significantly more effective than the parent compound in reducing the viability of prostate cancer cells and had the additional ability to cooperate with chemotherapy or immunotherapy. Saq-NO increased the transcription of the DR5 gene, making it more sensitive to TRAIL and thus significantly reducing cell viability. In addition, it enhanced the activity of another transcription factor, NF-κB, stimulating the expression of the DR5 enhancer CHOP, upregulating DR5 levels and inducing apoptosis in cancer cells.

Sulindac is the most widely studied chemoprophylactic non-steroidal anti-inflammatory drug with anti-inflammatory and anticancer activities [111]. Huang et al. studied the effect of sulindac on prostate cancer cells and found that sulindac inhibited the proliferation of and induced apoptosis of DU145 cells. They also demonstrated that sulindac induced apoptosis through triggering the DR5-dependent apoptosis pathway [112].

Orlistat is a novel inhibitor of fatty acid synthase [113]. Fujiwara et al. detected the effect of orlistat on the mRNA and protein expression of DR5 in DU145 and PC3 cells. The results showed that the mRNA and protein expression of DR5 was upregulated in a dose-dependent manner in DU145 and PC-3 cells. Further study of the mechanism of action revealed that upregulation of CHOP at mRNA and protein levels was observed in both cell lines with orlistat, but the activation of DR5 promoter was dependent on CHOP in DU145 cells but not in PC3 cells. In addition, orlistat induced reactive oxygen species (ROS), and ROS scavengers reduced sensitivity to TRAIL by inhibiting CHOP and DR5 expression in both cell lines. These results indicated that orlistat upregulated DR5 through two pathways, namely, the ROS-CHOP pathway and the ROS-direct pathway. In conclusion, orlistat induces cancer cell death by upregulating DR5 expression, and there are two different ways of upregulating DR5, namely, the ROS-CHOP pathway and the ROS-direct pathway [114].

## 5. Perspectives

Although circularly permuted TRAIL has been developed to the stage of marketing application in China, no DR5 agonist has previously been approved for marketing worldwide. DR5 agonists currently in clinical trials are mainly anti-DR5 antibodies and recombinant protein ligands for DR5, with no small-molecule compounds available at this time. However, research into small molecule DR5 agonists does not stop there. In this study, we found that some chemotherapeutic agents currently in clinical use also have a role in modulating DR5 in prostate cancer cells, a finding that may help expand the use of these agents. On the other hand, we have collected 38 small-molecule compounds that can induce apoptosis in prostate cancer cells through direct or indirect regulation of DR5, heralding the potential of these compounds for the treatment of prostate cancer. More research is needed on small-molecule drugs against prostate cancer by modulating DR5.

Our literature research revealed that six clinically used chemotherapeutic agents have been reported to promote DR5 expression. The six compounds are paclitaxel, vincristine, vinblastine, etoposide, adriamycin, and camptothecin (Figure 1, Table 1). Although these chemotherapy drugs are not first-line anti-prostate-cancer drugs, they are all spectrum antineoplastic agents. This finding suggests that there may be more antitumor drugs that modulate the DR5 signaling pathway in producing their antitumor effects and that how they drive the DR5 signaling pathway needs to be explored and studied. This finding also suggests that the combination of these compounds with DR5 agonists would theoretically lead to better clinical outcomes and may also reduce the dose of chemotherapeutic agents used, thereby reducing the adverse effects they cause. This is an area that deserves more research to explore and validate for application in the treatment of prostate cancer.

Similar to the chemotherapeutic agents mentioned above, none of the 28 natural compounds (Figure 2) we have mentioned in this article have been shown to be agonists of DR5; they all promote apoptosis in prostate cancer cells by increasing the expression of DR5 protein. Experiments with these compounds at the cellular level showed that all compounds promoted DR5 expression at the micromolar per liter concentration level, except for triptolide, which was at the nanomolar per liter concentration level (Table 2). Among these 28 compounds, diallyl trisulfide, Flavokawain B, and sulforaphane have been tested in animal studies to verify their anti-prostate-cancer effects. These studies suggest that natural products contain compounds that enhance the expression of DR5, and these compounds have been found to have anti-prostate-cancer effects at the cellular and/or animal levels and could be further investigated as anti-prostate-cancer drug lead compounds. In addition, we have collected 10 synthetic compounds (Figure 3) that can inhibit prostate cancer by increasing DR5 expression. These compounds were also dosed at micromoles per liter on cellular assays (Table 3). The effective concentrations were in the same magnitude as those of most of the natural products mentioned above. Two of the synthetic compounds, dihydroartemisinin and saquinavir-NO, have been shown to inhibit prostate cancer development in animals as well. Most promisingly, Nordihydroguaiaretic acid and Cyproterone acetate have progressed into phase Ⅱ and Ⅲ anti-prostate-cancer clinical trials, respectively (Table 4). Combining each of these two compounds with TRAIL may result in better clinical outcomes, while more studies need to be conducted to test this speculation.

Prostate cancer is mainly divided into androgen-dependent and androgen-independent types. In this review, we found that these small molecule compounds could induce prostate cancer cell apoptosis by enhancing DR5 expression in both androgen-dependent (LNCaP) and androgen-independent prostate cancer cells (PC-3, DU145). We therefore hypothesize that these compounds, which induce apoptosis in prostate cancer cells by increasing DR5 expression, are not clearly selective for androgen-dependent and androgen-independent prostate cancer cells. Some of the 38 compounds mentioned in this review have activity not only against prostate cancer, but also against other tumors. Additionally, studies of the anti-tumor effects of some of these compounds have been carried out to the clinical trial stage (Table 4). This indicates that these small molecule compounds can be tolerated by humans, and they hold great promise for development as new anti-cancer compounds.

In the present review, we summarize the drugs, natural compounds and synthetic compounds that can modulate DR5 expression and thus induce the apoptosis of prostate cancer cells (Figure 4). In terms of current research progress, the only small-molecule agonist of DR5 is bioymifi, which is also a synthetic compound. However, the role of bioymifi in prostate cancer has not yet been revealed. The 44 compounds mentioned in our review in Table 1, Table 2 and Table 3 are not agonists of DR5, but they can produce anti-prostate-cancer effects by promoting the expression of DR5. Smaller compounds have more stable physical and chemical properties than DR5 antibodies or ligand proteins. DR5 agonists or small-molecule sensitizers, which act specifically on the DR5 protein, can be used as an alternative to other chemotherapeutic agents after resistance has developed. These compounds need to be further investigated to clarify the specific molecular mechanisms by which they indirectly regulate DR5, so that new anti-prostate-cancer drugs or combination therapies can be developed.

## Figures and Tables

**Figure 1 pharmaceuticals-15-01029-f001:**
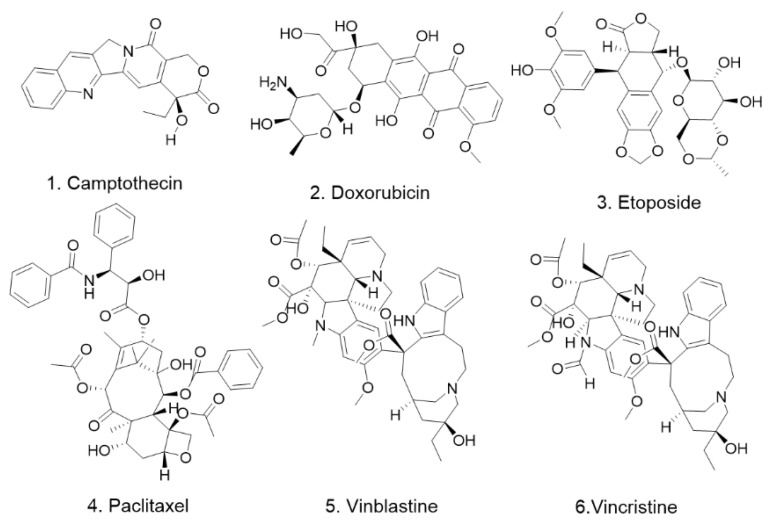
The chemical structures of compounds in Table 1.

**Figure 2 pharmaceuticals-15-01029-f002:**
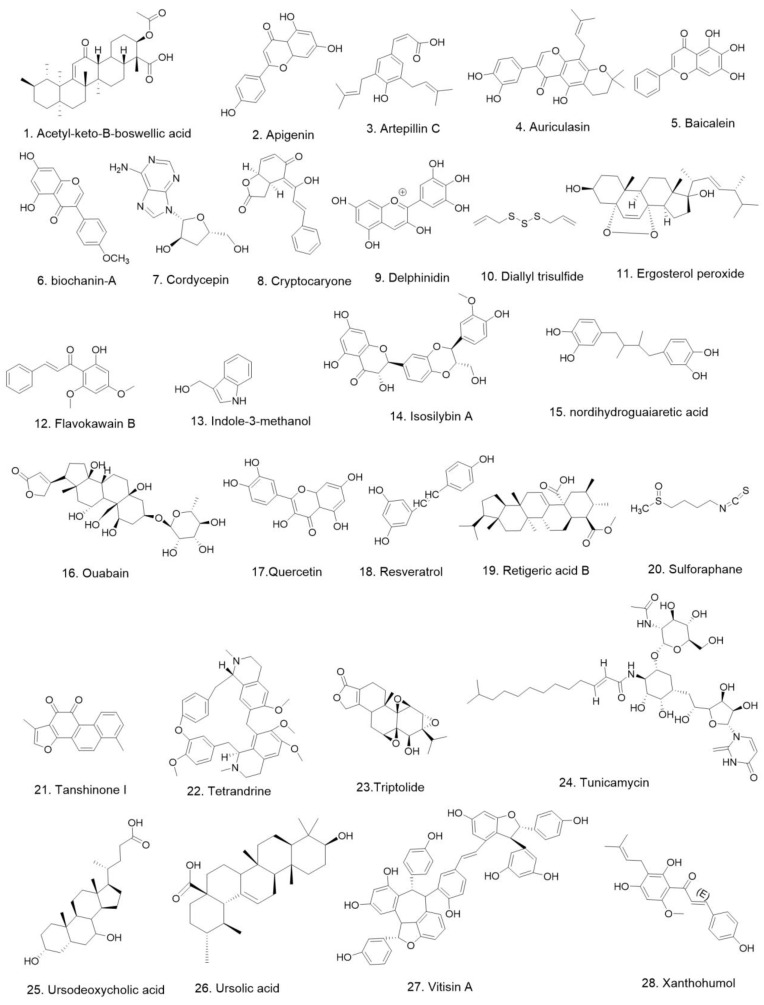
The chemical structures of compounds in Table 2.

**Figure 3 pharmaceuticals-15-01029-f003:**
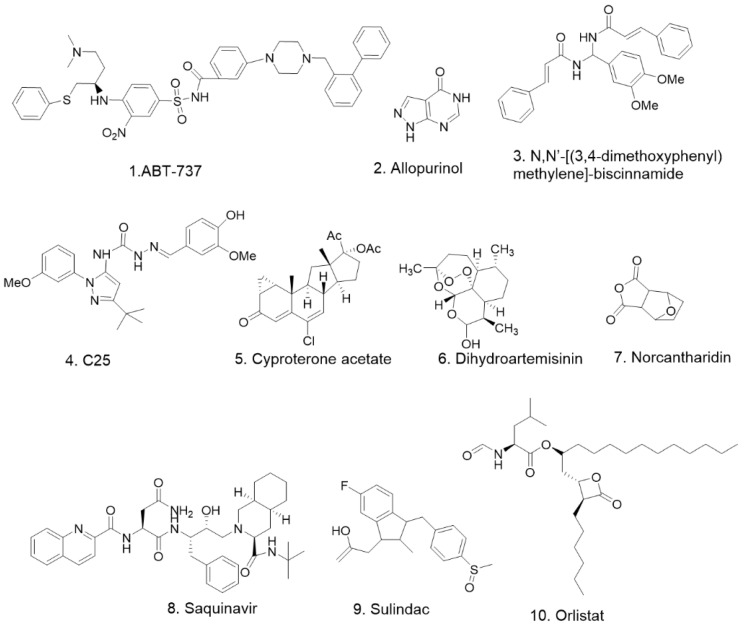
The chemical structures of compounds in Table 3.

**Figure 4 pharmaceuticals-15-01029-f004:**
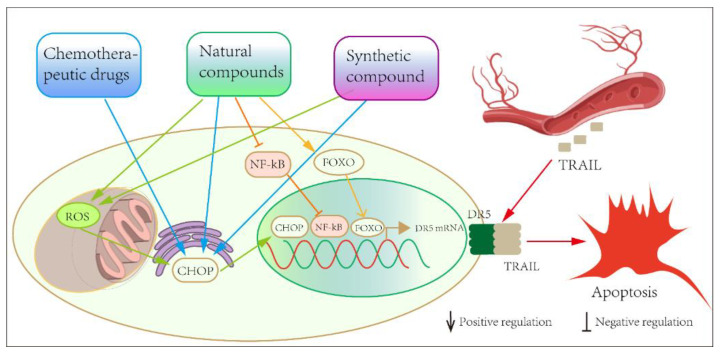
Overview of the mode of action of small-molecule compounds in regulating DR5-induced apoptosis.

**Table 1 pharmaceuticals-15-01029-t001:** Chemotherapeutic drugs suppress prostate cancer by enhancing the function of death receptor 5.

No.	Name	Source	Cell Line	Cell Experimental Concentration	Animal Experimental Concentration	Reference
1	Camptothecin	*Camptotheca acuminata* Decne	PC3/LNCaP/DU145	0 μM–50 μM	15 mg/kg (Balb/c nu/nu mice)	[17]
2	Doxorubicin	Boraginaceae	PC3/LNCaP/DU145	0 μM–50 μM	15 mg/kg (Balb/c nu/nu mice)	[17]
3	Etoposide	*Podophyllotoxin*	PC3/LNCaP/DU145	0 μM–50 μM	15 mg/kg (Balb/c nu/nu mice)	[17]
4	Paclitaxel	Pacific yew, Chinese yew	PC3/LNCaP/DU145	0 μM–50 μM	15 mg/kg (Balb/c nu/nu mice)	[17]
5	Vinblastine	*Madagascar rosy* periwinkle	PC3/LNCaP/DU145	0 μM–50 μM	N.D.	[17]
6	Vincristine	*Madagascar rosy periwinkle*	PC3/LNCaP/DU145	0 μM–50 μM	N.D.	[17]

Note: N.D: Not detected.

**Table 2 pharmaceuticals-15-01029-t002:** Natural compounds that suppress prostate cancer by promoting expression of death receptor 5.

No.	Name	Source	Cell Line	Cell Experimental Concentration	Animal Experimental Concentration	Reference
1	Acetyl-Keto-β-Boswellic Acid	*Boswellia serrata* and *Boswellia carterri* Birdw.	PC3/LNCaP	10 mg/mL–20 mg/mL	N.D.	[22]
2	Apigenin	Chamomile, honeybee, Perilla, verbena, yarrow	DU145/LNCap	5 μM, 10 μM, 20 μM	N.D.	[26]
3	Artepillin C	*Baccharis dracunculiforia*	LNCaP	50 μM–100 μM	N.D.	[29]
4	Auriculasin	*Flemingia philippinensis*	RWPE-1, RC-58T/h/SA#4	5 μM–10 μM	N.D.	[32]
5	Baicalein	Scutellaria baicalensis	PC-3	10 μM, 20 μM, 40 μM, 80 μM	N.D.	[35]
6	Biochanin-A	soy and red clover	LNCaP/DU145	20 μM, 50 μM, 100 μM	N.D.	[93]
7	Cordycepin	Cordyceps militaris	LNCap	20 μg/mL, 100 μg/mL, 150 μg/mL, 200 μg/mL	N.D.	[40]
8	Cryptocaryone	Cryptocarya infectoria	PC3/LNCaP/DU145	PC3, IC_50_ = 1.6 μM; DU145, IC_50_ = 2.3 μM; LNCaP, IC_50_ = 3.4 μM	N.D.	[41]
9	Delphinidin	fruits and vegetables	LNCaP/DU145	30 μM, 60 μM, 90 μM	N.D.	[43]
10	Diallyl trisulfide	garlic	PC3/LNCaP	10 μM–40 μM	40 mg/kg (BALB/c nu/nu mice)	[45]
11	Ergosterol peroxide	Sarcodon aspratus	DU 145	6.25 μM, 12.5 μM, 25μM, 50 μM	N.D.	[48]
12	Flavokawain B	Piper methystticum	LNCaP, LAPC4, DU145 and PC-3	1.1 μM, 2.2 μM, 4.4 μM, 8.8 μM, 17.6 μM	50 mg/kg	[50]
13	Indole-3-methanol	fruits and vegetables	LNCaP, DU145	30 μM, 60 μM, 90 μM	N.D.	[52]
14	Isosilybin A	*Silybum marianum*	LNCaP, LAPC4, 22Rv1	90 μM–180 μM	N.D.	[54]
15	Nordihydroguaiaretic acid	*larra triedentata*	DU145	2.5μM, 5 μM, 10 μM, 20 μM, 40 μM, 80 μM	N.D.	[55]
16	Ouabain	*Strophanthus gratus* and *Acocanthera ouabaio*	DU145	1.25 μM–40 μM	N.D.	[60]
17	Quercetin	*Bauhinia longifolia* (*Bong.*)	PC3/LNCaP/DU145/YPEN-1	10 μM–100 µM	N.D.	[61]
18	Resveratrol	grapes peanuts	PC3/DU145	0 μM–30 μM	N.D.	[63]
19	Retigeric acid B	*Lobaria kurokawae* Yoshim,	PC-3, DU145	2 μM, 4 μM, 6 μM, 8 μM and 10 µM	N.D.	[67]
20	Sulforaphane	Brassica oleracea italica	PC3/LNCaP	20 μM–40 μM	40 mg/kg (BALB/c nu/nu)	[69]
21	Tanshinone I	Salvia miltiorrhiza	PC3/DU145/M2182	20 μM, 40 μM, 80 μM	N.D.	[71]
22	Tetrandrine	*Stephania tetrandra*	LNCaP/PC3/RWPE-1	5 μM, 10 μM, 20 μM	N.D.	[73]
23	Triptolide	Tripterygium wilfordii	PC3/LNCaP/RWPE-2	50 nM–200 nM	N.D.	[76]
24	Tunicamycin	Streptomyces lysosuperficus	PC3/DU145	0.25 μg/mL, 0.5 µg/mL, 1 µg/mL, 2 µg/mL, 4 µg/mL	N.D.	[79]
25	Ursodeoxycholic acid	Bear bile	DU145	10 μg/mL, 20 μg/mL, 50 μg/mL, 100 μg/mL, 200 μg/mL	N.D.	[83]
26	Ursolic acid	Ligustrum lucidum Ait.	LNCaP, DU145, PC-3	10 μM, 20 μM, 30 μM, 40 μM	N.D.	[86]
27	Vitisin A	wine grapes	PC3/LNCaP/DU145	4 μM	N.D.	[87]
28	Xanthohumol	*Humulus lupulus L*	LNCaP	20 μM, 30 μM, 50 μM	N.D.	[90]

Note: N.D: Not detected.

**Table 3 pharmaceuticals-15-01029-t003:** Chemically synthesized compounds that suppress prostate cancer by enhancing the function of death receptor 5.

No.	Name	Cell Line	Cell Experimental Concentration	Animal Experimental Concentration	Reference
1	ABT-737	PC3/LNCaP	1 μM, 5 μM, 10 μM	N.D.	[93]
2	Allopurinol	PC3/DU145	12.5 μM, 25 μM, 50 μM, 200 μM	N.D.	[96]
3	N, N’-[(3,4-dimethoxyphenyl) methylene]-biscinnamide	PC-3	10 μM, 30 μM	N.D.	[99]
4	C25	LNCaP	10 μM, 15 μM	N.D.	[99]
5	Cyproterone acetate	HEK293/PC3/DU145	50 μM	N.D.	[102]
6	Dihydroartemisinin	PC3/LNCaP/DU145	10 μM, 30 μM, 50 μM	100 mg/kg (mouse)	[105]
7	Norcantharidin	22Rv1/DU145	3 μg/mL, 10μg/mL, 30 μg/mL	N.D.	[107]
8	Saquinavir-NO	PC3	4.7 μM, 9.4 μM, 18.8 μM	0.2 mg/mouse (BALB/c female athymic nude mice)	[110]
9	Sulindac	DU145	200 μM	N.D.	[112]
10	Orlistat	DU145 and PC3	25 μM, 50 μM, 100 μM, 200 μM	N.D.	[114]

Note: N.D: Not detected.

**Table 4 pharmaceuticals-15-01029-t004:** The small molecule compounds in this paper that have been tested in anti-tumor clinical trials.

No.	Name	Cancer	Phase	Reference
1	ABT-737	ovarian Cancer	Ex Vivo	https://clinicaltrials.gov/ct2/show/NCT01440504?term=ABT-737&cond=cancer&draw=2&rank=1 (accessed on 15 August 2022)
2	cordycepin	advanced cancers, lymphomas, solid tumors, and bone marrow tumors	I/II	[115]
3	Indole-3-carbinol	breast cancer	I	[116]
4	Nordihydroguaiaretic acid	prostate cancer	II	[58]
5	Quercetin	oral cancer	II	[117]
6	Resveratrol	colon cancer and liver cancer	I/II	[118]
7	Sulforaphane	bladder and prostate cancer and breast cancer	II	[119]
8	Triptolide	solid tumors	I (Recruiting)	https://clinicaltrials.gov/ct2/show/NCT05166616?term=Triptolide&cond=cancer&draw=2&rank=1 (accessed on 15 August 2022)
9	Ursodeoxycholic acid	duodenal tumors	III	[120]
10	Allopurin	small cell tumors	I	[121]
11	Cyproterone acetate	prostate cancer	III	[103]
12	Norcantharidin	solid tumors	I (Recruiting)	https://www.clinicaltrials.gov/ct2/show/NCT04673396?term=Norcantharidin&draw=2&rank=1 (accessed on 15 August 2022)
13	Sulindal	colorectal, breast, and thyroid-free cancers	II	[122]

## Data Availability

Data sharing not applicable.

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
