# Peer review of "Pharmacological Small Molecules against Prostate Cancer by Enhancing Function of Death Receptor 5"

_pharmaceuticals, 2022, doi:10.3390/ph15081029_

Round 1

Reviewer 1 Report

The authors have done a comprehensive literature search on studies that have focused on using compounds that target DR5 for prostate cancer and have shown that some of these could have the potential to be effective therapeutic options for prostate cancer by induction of apoptosis selectively in cancer cells.

Suggestions and questions to the authors:

1. In the introduction, the authors wrote - We screened and excluded irrelevant papers by article titles and abstracts and obtained 32 papers that met the requirements. Could the authors expand on why they excluded based on just title and abstracts since there could still be information on apoptosis in the article which is not highlighted in the abstract. Would the authors acknowledge that they might have missed some studies due to this screening criteria? Also, when the authors say that 32 papers meet the requirements it would be important to write those requirements in details for the readers. 

2. In  my opinion, the introduction needs more information on DR5 and the extrinsic apoptosis pathway and how it induces apoptosis to help the readers understand why increase in its expression could be useful for targeting purposes. 

3. Please include this study or any related study in introduction -https://www.ncbi.nlm.nih.gov/pmc/articles/PMC4214341/ where they show DR5 expression correlation in prostate cancer to show significance of this literature search for DR5 targets.

4. Even though this study is limited to prostate cancer but it would be nice to mention a paragraph on the compounds if they have been used for any trial studies in any other cancer type as that could provide some hints into if they could be tolerated by humans or not?

5. Please provide a small paragraph of the best possible combination therapies that would be effective along with DR5 targeting compounds.

6. DR5 antibodies are being also tested for certain cancers. Please mention that as a potential option that needs to be tested in prostate cancer model if it has not been done yet.

Author Response

Dear reviewer,

Thank you for your comments concerning our manuscript entitled “Pharmacological small molecules against prostate cancer by enhancing function of death receptor 5”. (ID: pharmaceuticals-1867791). Those comments are all valuable and very helpful for us to revise and improve our paper. We studied comments carefully and have made related corrections in the revised manuscript.

The responses to the comments are listed in the response file. Please see the attachment.

Reviewer 2 Report

The authors described the potential role of DR5 in human prostate cancers and arranged small molecules that can modulate DR5. In addition, they have schematically illustrated how these small molecules can target DR5. Although the manuscript is informative, some minor points should be addressed as follows.

1. Prostate cancers are divided mainly into androgen-dependent or independent, so the authors should discuss whether DR5 has a different function in both classes of prostate cancers.

2. The authors should discuss more what advantages small molecules targeting DR5 might have over antibodies or proteins.

3. The biggest problem with CRPC is that it frequently develops resistance to the drug administered. I suggest authors discuss whether resistance can be overcome by targeting DR5 and whether such resistance is unlikely to occur by targeting DR5.

4. Typological errors should be carefully checked and corrected.

Author Response

(The authors gave the same response as above.)

Round 2

Reviewer 1 Report

The authors have addressed all my concerns very well. The manuscript can be accepted.